# Is Gender an Important Factor in the Precision Medicine Approach to Levocetirizine?

**DOI:** 10.3390/pharmaceutics16010146

**Published:** 2024-01-21

**Authors:** Seung-Hyun Jeong, Ji-Hun Jang, Yong-Bok Lee

**Affiliations:** 1College of Pharmacy, Sunchon National University, 255 Jungang-ro, Suncheon-Si 57922, Jeollanam-do, Republic of Korea; jeongsh@scnu.ac.kr (S.-H.J.); jangji0121@naver.com (J.-H.J.); 2College of Pharmacy and Research Institute of Life and Pharmaceutical Sciences, Sunchon National University, Suncheon-Si 57922, Republic of Korea; 3College of Pharmacy, Chonnam National University, 77 Yongbong-ro, Buk-gu, Gwangju 61186, Republic of Korea

**Keywords:** gender differences, levocetirizine, pharmacodynamics, population pharmacokinetic modeling, precision medicine, covariates

## Abstract

Currently, there is insufficient information on the variability in levocetirizine pharmacometrics among individuals, a crucial aspect for establishing its clinical use. The gender differences in pharmacokinetics and the extent of variation in pharmacodynamics have not been definitively identified. The primary goal of this study was to investigate gender differences in levocetirizine pharmacokinetics and quantitatively predict and compare how these gender-related pharmacokinetic differences impact pharmacodynamics, using population pharmacokinetic–pharmacodynamic modeling. Bioequivalence results for levocetirizine (only from the control formulation) were obtained from both healthy Korean men and women. Physiological and biochemical parameters for each individual were utilized as pharmacokinetic comparison and modeling data between genders. Pharmacodynamic modeling was performed using reported data on antihistamine responses following levocetirizine exposure. Gender, weight, body surface area, peripheral distribution volume, albumin, central–peripheral inter-compartmental clearance, and the fifth sequential absorption rate constant were explored as effective covariates. A comparison of the model simulation results showed a higher maximum concentration and faster plasma loss in females than in males, resulting in a faster recovery to baseline of the antihistamine effect; however, the absolute differences between genders in the mean values were not large within 10 ng/mL (for plasma concentrations) or % (wheal and flare size changes). Regarding the pharmacokinetics and pharmacodynamics of levocetirizine, the gender effect may not be significant when applying the usual dosage (5 mg/day). This study will be useful for bridging the knowledge gap in scientific precision medicine by introducing previously unconfirmed information regarding gender differences in levocetirizine pharmacometrics.

## 1. Introduction

Levocetirizine is an antihistamine drug used to relieve symptoms of seasonal or perennial allergic rhinitis in adults and children over 6 years of age [1,2]. It is also used to treat chronic idiopathic urticaria, dermatitis with itching, and eczema [3]. Pharmacologically, levocetirizine acts through a highly selective antagonism of histamine type I receptors [4], resulting in the inhibition of several responses such as the histamine-induced increase in vascular endothelial permeability, stimulation of cough receptors, and stimulation of the wheal and flare responses in the nervous system [5]. Levocetirizine is chemically and structurally a piperazine derivative that is an oxidized metabolite of hydroxyzine, a first-generation antihistamine [6]. Unlike cetirizine, which is stereoisomerically a racemic mixture, levocetirizine is in a highly pure R-form [7]. Levocetirizine is a second-generation antihistamine drug and, unlike first-generation antihistamines, has less transfer to the central nervous system and thus less side effects such as drowsiness [4,8]. Therefore, when selecting a drug in clinical practice for the purpose of inducing an antihistamine effect, levocetirizine is a highly preferred drug in terms of side effects and patient compliance. Despite the frequent clinical application of levocetirizine [9], there is still a lack of comprehensive information regarding the quantitative pharmacokinetics and pharmacodynamics of levocetirizine in individuals. In other words, the pharmacometrics data available for setting scientific usage and predicting results are restricted to a handful of elements, indicating a notable knowledge gap in integrated pharmacometric analysis.

In particular, access to quantitative information about the variations in the pharmacokinetics and pharmacodynamics of levocetirizine between genders has been very limited. It is generally accepted that inherent physiological differences between genders [10] can serve as a substantial factor, leading to significant differences in pharmacokinetics when exposed to drug formulations with identical content. Consequently, these differences may contribute to variations in pharmacodynamics [11]. Therefore, considering the inevitable need to consider gender factors in clinical application, it was very urgent and important to conduct research focusing on exploring the degree of differences between genders in the pharmacometrics of levocetirizine. However, verifying differences in pharmacokinetics between genders faces challenges, including the need for a gender-inclusive clinical trial design and the complexity of interpreting comprehensive results. Therefore, scientific reports detailing pharmacometric differences between genders remain notably scarce. The primary objective of this study was to investigate previously unrecognized gender differences in levocetirizine pharmacokinetics within the population and to quantitatively interpret the degree of pharmacodynamic differences resulting from pharmacokinetic differences. In addition, we sought to discover and interpret the pharmacokinetic relationships of physiological and biochemical factors that are effective in revealing the inter-individual pharmacokinetic diversity of levocetirizine. The assessment of efficacy and safety, derived from interpreting the pharmacokinetic differences of levocetirizine based on gender and individual factors, is garnering significant clinical interest. Additionally, the discovery of quantitative predictors related to pharmacokinetic diversity will be a progressive process that accelerates precision medicine. The results of the gender-specific pharmacokinetic analysis and pharmacometrics modeling of levocetirizine presented in this study provide valuable data for evidence-based clinical treatment and the progression of research that accounts for inter-individual variability (IIV) of levocetirizine.

## 2. Materials and Methods

### 2.1. Research Approach

This research involved five key stages, and the schematic diagram is presented in Appendix A. In the first step, pharmacokinetic comparisons between genders were executed through non-compartmental analysis (NCA) and graphical profiling. The data for these comparisons were obtained from bioequivalence results, with gender taken into account from the clinical design stage. And, the results obtained only from the control formulation, not the test formulation, were used in this pharmacokinetic comparative study. Second, we conducted a correlation analysis between the pharmacokinetic parameters and the physiological and biochemical values of each individual. This proactive screening aimed to identify potential covariates relevant to interpreting the pharmacokinetic diversity of levocetirizine within the population. Subsequently, in the third step, population modeling was executed and validated using the pharmacokinetic data for levocetirizine. During this process, a basic model structure was established, incorporating the specification of intra- and inter-individual error models, and a step-by-step reflection of covariates to explain the pharmacokinetic diversity between individuals was conducted. Fourth, pharmacodynamic modeling was performed based on reports of plasma concentrations and antihistamine responses following exposure to levocetirizine. Additionally, the pharmacodynamic model developed in this study was integrated with the pharmacokinetic model, enabling the simulation of drug efficacy based on variations in levocetirizine concentration in plasma. In the fifth step, quantitative prediction simulations of levocetirizine’s pharmacokinetics and efficacy were conducted, taking into account relevant covariates, with a specific focus on gender, in the established pharmacometrics model of levocetirizine.

### 2.2. Pharmacokinetic Comparison between Genders

A pharmacokinetic comparison between genders was performed based on the results of a clinical study of levocetirizine 5 mg tablets (UCB Pharma, Brussel, Belgium; lot number: 319826) conducted on 24 healthy Korean men and 16 women. The clinical trial was completed with a total of 40 participants (who did not experience side effects), and their demographic information (including physiological and biochemical parameters) is presented in Appendix A. Inclusion criteria for the participants were as follows: age of 19 years or older at screening; obesity index BMI falls within the range of 18–30 kg/m^2^; weight greater than 50 kg and 45 kg for men and women, respectively; no clinically significant congenital or chronic diseases, and no pathological symptoms or findings as a result of a medical examination; and diagnostic tests (blood and urine) and electrocardiogram results were within normal values. The exclusion criteria were as follows: taking drugs that induce or inhibit drug-metabolizing enzymes, such as barbiturates, within 30 days before the test date, or taking drugs that are likely to affect the test within 10 days; having a history of gastrointestinal resection, which may affect drug absorption; excessive drinking within 1 month prior to the test date; hypersensitivity to the test drug or its components (especially hydroxyzine and piperazine types); suspected pregnancy or pregnant and lactating; and a clinical history of mental illness. In addition, those who were on their menstrual period (for women) and/or were planning to become pregnant within 1 month (for both men and women) were additionally excluded from this clinical trial. A biochemical parameter analysis was conducted on all participants involved in the clinical trial, and the methods are detailed in Appendix A provides details on the selection and progression of the clinical trial participants, while Appendix A includes information on the clinical trial design and sampling. The clinical trial protocol underwent a comprehensive review and received official approval (approval number: MB22-013; approved on 25 March 2022) from the Ministry of Food and Drug Safety (Cheongju-si, Republic of Korea). A pharmacokinetic comparison between genders was performed by comparing the significance of the levocetirizine pharmacokinetic parameters for males and females calculated via NCA. The pharmacokinetic parameters, determined through NCA, were assessed using plasma concentration values over time, following the oral administration of levocetirizine 5 mg tablets. Appendix A briefly outlines the analytical method used to quantify levocetirizine in the plasma samples, and comprehensive methods for calculating the pharmacokinetic parameters through NCA are provided in Appendix A.

Additional NCA analyses were performed based on the normalized levocetirizine plasma concentration values for body weight and body surface area (BSA), which are major physiological differences between genders, and the results were categorized and compared by gender. Here, the methods for calculating pharmacokinetic parameters using NCA were the same as stated in Appendix A. A comparison of significant differences between genders was performed using a two-tailed *t* test, and the significance of all statistical tests was determined at a *p* value of 0.05. A graphical comparison of the levocetirizine plasma concentration (including normalized values for body weight and BSA) profiles by gender further confirmed differences in the pharmacokinetics between genders.

### 2.3. Population Pharmacokinetic Modeling

The development and evaluation of the population pharmacokinetic model for levocetirizine were executed employing a non-linear mixed effects model approach with Phoenix NLME software (version 8.4, Certara Inc., Princeton, NJ, USA). The estimation of population pharmacokinetic parameters for levocetirizine utilized the first-order conditional estimation method with extended least squares estimation, including the η–ε interaction. The construction of the population pharmacokinetic model involved two main steps. Initially, the development of a basic structural model was conducted to elucidate the plasma concentration of levocetirizine following its oral administration. This procedure involved ascertaining the number of basic compartments, considering the lag time (T_lag_) in the oral absorption of levocetirizine, implementing mechanistic structuring (including multiple absorption) in the absorption process, and selecting error models to accommodate residual and IIV of levocetirizine. During the model establishment, the values derived from the NCA were utilized as the initial parameter values, facilitating expedited parameter convergence and ensuring effective modeling. In the second step, a systematic search was conducted to identify relevant and significant covariates for modeling the IIVs in levocetirizine pharmacokinetics. This process involved the sequential application of candidate covariates to the IIV model of the pharmacokinetic parameters embedded in the model. As potential effective covariates, the gender element and the physiological and biochemical factors of the individuals obtained in this study were all considered, with each treated as categorical (for the gender factor) and continuous data (for the physiological and biochemical factors). The selection of suitable models for each step utilized diverse statistical significance tools provided by Phoenix NLME. This procedure involved computing twice the negative log likelihood (−2LL), Akaike’s information criterion (AIC), and goodness-of-fit (GOF) plots. Additionally, the significance of the total number of parameters applied to the model (change in degrees of freedom) was taken into account. The significance was assessed using a chi-square distribution *p* value of 0.05 for forward selection and 0.01 for backward elimination, both at −2LL and the objective function value (OFV). The adequacy of the developed population pharmacokinetic model for levocetirizine was confirmed through GOF analysis, which included an examination of residual distribution, as well as through a visual predictive check (VPC) and bootstrapping processes. Detailed approaches for each verification tool are provided in Appendix A.

### 2.4. Expansion to a Pharmacodynamic Model

The pharmacodynamic model was constructed using measured information [12] on the plasma levocetirizine concentration and drug efficacy according to levocetirizine exposure. The medicinal efficacy of levocetirizine was indicated by changes in the wheal and flare sizes induced by exogenous histamine, which was related to the appearance of an anti-allergic response produced by levocetirizine in the plasma selectively inhibiting the histamine type I receptors [12]. The pharmacodynamic prediction with levocetirizine was made possible by structuring the effect compartment to be related to the pharmacokinetic profile of levocetirizine in the central compartment. In other words, alterations in the pharmacokinetic profile in plasma over time following exposure to levocetirizine directly influence changes in drug efficacy. This relationship could be extended and translated into a pharmacodynamic model, allowing for the quantification of the time–drug effect after exposure to levocetirizine. This was structurally consistent with the attempted areas in previous pharmacokinetic and pharmacodynamic co-linkage models [13,14,15,16]. 

The data used in this study to attempt to expand the levocetirizine pharmacodynamic model were the pharmacokinetic and pharmacodynamic results for a population ranging from 6 to 11 years of age [12]. However, in the final conclusion of the report [12], the appropriate levocetirizine dosage was suggested to be 5 mg once daily as in adults, which was derived based on clinical pharmacokinetic and pharmacodynamic studies. Therefore, even though these data were from individuals in the 6 to 11 years age group, the data were judged to be sufficiently applicable in attempting to expand the pharmacodynamic model of this study. This was because factors that could lead to a significant difference in the pharmacokinetic–pharmacodynamic correlation between the 6–11 years age group and the adult group were not identified. Additionally, the report was the only realistically accessible data in which the clinical pharmacokinetic–pharmacodynamic profiles of levocetirizine were clearly presented [12]. 

In the extension to the pharmacodynamic model, pharmacokinetic–pharmacodynamic data for the 6 to 11 years age group [12] were applied to the population pharmacokinetic model structure established in this study, enabling the structuring of the pharmacodynamic model and an estimation of the parameter values. Additionally, the established pharmacodynamic model structure was extended and applied to the population pharmacokinetic model of levocetirizine developed in this study. The co-linkage analysis technique of the internal pharmacokinetic dataset-based model and the external pharmacodynamic dataset-based model of levocetirizine based on rational judgment, which was expanded and applied in this study, was similar to the methodological part attempted in past reports [13,14]. Since the pharmacokinetic and pharmacodynamic patterns of the reported data [12] were sufficiently model-fitted at a reasonable level, no major problems were foreseen in the attempt to expand the pharmacodynamic modeling in this study if significant correlation differences between pharmacokinetics and pharmacodynamics did not occur according to age factors.

### 2.5. Model Simulation

The model simulation was conducted to both qualitatively and quantitatively assess alterations in the pharmacokinetic profile. This process considered the covariates explored in this study, especially gender. This simulation aimed to confirm the resulting pharmacodynamic effects. In conducting pharmacodynamic simulations with the population pharmacokinetic model, the model structure established and validated in this study was kept constant. Moreover, the parameter values of the model were set to the mean values of the group, excluding parameters considered to reflect covariates. The parameters affected by covariates were set as the mean values of the group, adjusting for changes in covariate values. Additionally, the correlation between covariates and parameters was taken into consideration. Model simulations were performed to predict and compare the resulting pharmacokinetic–pharmacodynamic patterns with changes in pharmacokinetic parameter values reflecting the covariates. The models were simulated using the simulation and prediction engines of Phoenix NLME.

## 3. Results

### 3.1. Pharmacokinetic Differences between Genders

Plasma concentration profiles, plotted against time, following a single oral administration of 5 mg levocetirizine tablets are presented in Figure 1. Oral absorption of levocetirizine began 0.17 h after administration and progressed rapidly to approximately 2 h. This was thought to be related to the characteristics of the immediate release (IR) formulation (having rapid disintegration and dissolution in the gastrointestinal tract) and the rapid permeation of levocetirizine in the gastrointestinal tract [17]. As a result of comparing the levocetirizine pharmacokinetic profiles between genders, higher plasma concentration values were found in women than in men in terms of absorption and distribution (approximately 1 to 6 h after oral administration), and the after-distribution and elimination phases (approximately 24 to 36 h after oral administration) were higher in men. The pharmacokinetic profile results of plasma concentrations normalized to body weight and BSA, respectively, were similar to the pharmacokinetic profile results based on raw data. In other words, levocetirizine absorption and distribution showed higher plasma concentration values in women than in men, while the post-distribution and elimination phases were higher in males.

Body weight and BSA are obvious physiological differences that inevitably occur between genders. In comparing the physiological and biochemical factors of the gender groups participating in this clinical study (Appendix A), significant differences (*p* < 0.05) in body weight and BSA values were confirmed. On the other hand, although gender differences in such covariates as glomerular filtration capacity, liver function indices, and albumin were confirmed in this study (Appendix A), there were limitations in clarifying whether the differences were directly attributable to gender factors. Therefore, the plasma concentration raw data were normalized to body weight and BSA, which corresponded to clear and reasonable physiological differences between genders, respectively. Additional profile and NCA analyses were performed to more clearly compare and interpret the differences in pharmacokinetics between genders by removing the basic influencing factors. In addition, several reports [18,19,20] have shown that body weight and BSA can be effective covariates that affect the changes in pharmacokinetic parameters, including the clearance (CL/F), volume of the distribution (V/F), and maximum plasma concentration (C_max_) of a drug. 

Appendix A shows the pharmacokinetic parameter values of levocetirizine obtained through NCA based on the raw plasma concentration data following oral administration of levocetirizine 5 mg tablets. The NCA results for all subjects showed that the mean time to reach C_max_ (T_max_) was 1.11 h, suggesting rapid oral absorption of levocetirizine, consistent with the pharmacokinetic profiles in Figure 1. The mean V/F was 33.14 L, suggesting a wide distribution of levocetirizine in the body, including plasma, inter-tissue fluid, and intra-tissue fluid. The mean levocetirizine half-life (T_1/2_) was 8.06 h, suggesting that approximately 90% of the levocetirizine administered as a single oral dose would be eliminated within 24 h. This indirectly implied that levocetirizine 5 mg tablets could be safely used without significant accumulation in the body when administered once a day or once every 2 days. A comparison of the NCA results between genders showed that the C_max_ of levocetirizine was significantly higher, and the T_1/2_ and mean residence time (MRT) were significantly shorter in women than in men (*p* < 0.05). Additionally, the V/F of levocetirizine was significantly lower in women than in men (*p* < 0.05). There were also no significant differences between genders in the area under the curve (AUC), CL/F, and T_max_ of levocetirizine (*p* > 0.05). This suggests that the maximum degree of oral absorption of levocetirizine is significantly lower, the retention time in plasma is longer, and the distribution in the body is wider in males than in females. On the other hand, it was implied that there would be no significant difference between genders in terms of total exposure in plasma and the degree of elimination in the body after levocetirizine administration. Appendix A show the pharmacokinetic parameter values calculated through the NCA process based on the levocetirizine plasma concentrations normalized to the individual body weight and BSA, respectively. In these parameter values, all except T_max_ showed significant differences between genders (*p* < 0.05). The AUC and C_max_ tended to be higher, and the CL/F and V/F were lower in females than in males. Based on gender factors excluding body weight and BSA, it was implied that levocetirizine exposure and maximum absorption in plasma were higher in women than in men, while the degree of distribution and elimination in the body was lower. When comparing the NCA results calculated based on the levocetirizine raw plasma concentration data (Appendix A), common gender significance was confirmed in the C_max_ and V/F. This suggests that, in addition to body weight and BSA factors, the gender factor affects the levocetirizine concentration in plasma and degree of distribution in the body. As a result, gender factors as well as weight and BSA factors cannot be completely ignored when considering changes in the plasma concentration following exposure to the same dose of levocetirizine tablets.

### 3.2. Levocetirizine Population Pharmacokinetic Model

The population pharmacokinetics of levocetirizine were described using the five sequential model with T_lag_ for gastrointestinal absorption and the distribution by the central-to-peripheral two-compartment model. Appendix A presents the structure of the population pharmacometrics model of levocetirizine established in this study. As for the basic compartment, significant model improvement was confirmed in the two-compartment rather than one-compartment model (−2LL reduction, *p* < 0.05 and/or 0.01). In structures with three compartments or more, the reduction in −2LL was not statistically significant with the increase in the total number of parameters. Therefore, the plasma concentration profiles of levocetirizine could be characterized by its distribution within both the central and peripheral compartments, exhibiting two kinetics in the body. In examining the levocetirizine absorption phase, various structural models were explored. These included the T_lag_ reflection model, the non-sequential multiple absorption model (involving two or more absorption points with consideration of bioavailability), and the sequential multi-compartment absorption model (applying two or more absorption rate constant parameters between successive absorption compartments). Additionally, various mathematical transformation models, including the Weibull absorption, saturation, zero-order, and mean transit time (MTT), were explored. As a result, the five sequential first-order absorption with T_lag_, which showed the largest −2LL reduction, was selected as the most appropriate model to explain the absorption pattern of levocetirizine. Other structural absorption models, excluding zero-order, exhibited a reduction in −2LL when compared to the basic model (absence of T_lag_ with first-order). However, the extent was relatively less than that observed in the sequential first-order multi-compartmental absorption model with T_lag_. The significance of the sequential absorption compartments was observed up to five (*p* < 0.01) while significantly improving the GOF plots compared to the basic model. However, with six or more compartments, −2LL increased as the total number of parameters rose. The log additive error model was suitable as a residual error model, and when applied, the overall number of parameters was maintained and the degree of reduction in −2LL was very high at 124.45%. Furthermore, the utilization of residual error models, including additive, power, and mixed error, led to a significant elevation in −2LL compared to the proportional error used in the basic model. The IIVs in the pharmacokinetic parameters of levocetirizine were clarified through the application of an exponential error model. A step-by-step identification of the need for IIV for model improvement led to the inclusion of IIVs for all parameters described in the model. When the IIV consideration was sequentially excluded for each parameter of the central compartment distribution volume (V_c_/F), central compartment clearance (CL_c_/F), peripheral compartment distribution volume (V_p_/F), central-to-peripheral inter-compartment clearance (CL_p_/F), first absorption rate constant (K_a1_, dosing depot to depot 1), second absorption rate constant (K_a2_, depot 1 to depot 2), third absorption rate constant (K_a3_, depot 2 to depot 3), fourth absorption rate constant (K_a4_, depot 3 to depot 4), fifth absorption rate constant (K_a5_, depot 4 to central compartment), and lag time (T_lag_), the −2LL increased beyond a significant value (*p* < 0.05 and/or 0.01) in all cases. An assessment of the need for IIV of the parameters was carried out by evaluating the extent of model improvement through a stepwise elimination of the IIV for each parameter. This process was performed based on the full model, where all the IIVs of the model parameters were initially considered. Appendix A provides an overview of the steps taken to establish the basic pharmacokinetic structural model for levocetirizine. A comprehensive evaluation of several physiological and biochemical factors measured during the clinical trials, along with gender factors, was conducted to identify candidate covariates capable of explaining the inter-individual pharmacokinetic variation in levocetirizine. Stepwise addition and elimination processes were used in an attempt to reflect the candidate covariates in the basic structural model. The aim was to find a correlation model with a significant OFV change by sequentially adding or removing candidate covariates to or from model parameters, considering IIV. A significant correlation was confirmed in all steps of both forward selection and backward elimination for model parameters, with standard *p* values (for statistical verification) of 0.05 and 0.01, respectively. Finally, BSA, albumin, and gender factors were correlated with V_p_/F as effective covariates in the population pharmacokinetic model of levocetirizine. The BSA factor was explored as an effective covariate for CL_p_/F, and body weight was explored as an effective covariate for K_a5_. As a result, the population pharmacokinetic variations in levocetirizine could be explained in relation to the distribution of levocetirizine in peripheral tissues, clearance between central and peripheral compartments, and variations in the rate of levocetirizine absorption into the central compartment. Appendix A summarizes the results showing a significant reduction in the OFV (*p* < 0.05 and/or 0.01) following the sequential application of candidate covariates to the established basic population pharmacokinetic model parameters for levocetirizine. The structural equations for the finally established population pharmacokinetic model of levocetirizine are outlined in Appendix A, while Appendix A contains the model parameter values and relevant data. The coefficient of variation (CV) of the typical values of the pharmacokinetic parameters V_c_/F, CL_c_/F, T_lag_, V_p_/F, CL_p_/F, K_a1_, K_a2_, K_a3_, K_a4_, and K_a5_ was within a reasonable range of 30% (Appendix A). Additionally, the CV of all parameters (including the covariate correlation values and intra- and inter-subject variability distribution values) were within the acceptable range of 50%. The higher estimates of 28.12 L and 13.80 L/h in the V_p_/F and CL_p_/F compared with 1.07 L and 2.94 L/h in the V_c_/F and CL_c_/F, respectively, suggested widespread body (especially peripheral tissues) distribution and a significant CL of levocetirizine. According to the covariate correlation results, a significantly negative relationship was confirmed between the V_p_/F and albumin, which was related to the very high (over 90%) plasma protein binding rate of levocetirizine and its reported ability to carry albumin in the blood. In other words, as the albumin level increases, the increased binding of levocetirizine to plasma proteins may limit its distribution to peripheral tissues. Positive covariate correlations between the V_p_/F and CL_p_/F and BSA suggested that as the BSA increased, levocetirizine’s peripheral tissue distribution and inter-compartmental (central to peripheral) CL increased. The positive covariate correlation between the K_a5_ and body weight suggested that the plasma absorption of levocetirizine would occur more rapidly as body weight increased. On the other hand, the negative covariate correlation between categorized gender and the V_p_/F implied that levocetirizine distribution in the peripheral tissues was lower in women than in men, which was consistent with the pattern confirmed in the NCA. 

The validation results of the established levocetirizine population pharmacokinetic model were all confirmed to fall within a reasonable range. Because of the bootstrap, the median values of all parameters fell within the 95% confidence interval, and the typical value estimates of the parameters in the final model did not differ largely (about 20%) from the bootstrap median values (Appendix A). Bootstrapping results confirmed the robustness and reproducibility of the final population pharmacokinetic model of levocetirizine. Additionally, no notable problems were identified in the GOF plot results. Appendix A shows the GOF plot results of the levocetirizine population pharmacokinetic model established in this study. It was confirmed that more than 90% of the total conditional weighted residuals (CWRES) according to time and model predictions fell within a reasonable range (within ±4 of 0). In addition, the distribution of the CWRES was normal and symmetrical with respect to 0. Individual and model predictions and observations showed high proportional agreement with each other, and the quantile–quantile (QQ) plot also showed an almost linear correlation. Some CWRES values deviating from ±4 may have been related to the high inter-individual absorption variability of levocetirizine. Because of the model VPC, it was confirmed that levocetirizine was lost from plasma relatively faster in the female group than in the male group and that the C_max_ was higher overall. This was the same result as the pharmacokinetic profile analysis based on the raw data (Figure 1). It was also confirmed that all observations (for the total data without stratification by gender and the stratified data) were well within the 95% VPC regions of the 97.5th, 50th, and 2.5th percentiles of the model predictions. Above all, it was confirmed that most observations were symmetrically distributed based on the 50th percentiles of the model predictions. As a result, it was implied that the levocetirizine population pharmacokinetic model established in this study explained the differences in levocetirizine pharmacokinetics between genders at a reasonable level. Figure 2 shows the VPC results of the levocetirizine population pharmacokinetic model. Because of the individual model fit check for each subject, the levocetirizine pharmacokinetic profiles of all subjects were explained well by the levocetirizine population pharmacokinetic model established in this study. The model predictions were fitted appropriately without significant bias from the observations. This again suggested the adequacy and predictive excellence of the levocetirizine pharmacokinetic model established in this study. Appendix A shows the comparison results between the model predictions and observations for each subject produced via the levocetirizine population pharmacokinetic model.

### 3.3. Pharmacodynamic Model

The structure and parameter values of the established and validated population pharmacokinetic model for levocetirizine were set as representative values for the population. Subsequently, these values were utilized to extend the model for predicting the pharmacodynamics of levocetirizine. This was conducted to explore the impact of the inter-individual pharmacokinetic variations due to gender, body weight, BSA, and albumin on drug efficacy. The pharmacodynamic model simulation was structured to co-link with changes in the levocetirizine plasma concentration pattern in the central compartment so that the antihistamine response caused by levocetirizine would appear in response to the levocetirizine concentration in the plasma. Changes in the wheal and flare sizes induced by exogenous histamine were established as pharmacodynamic indicators of levocetirizine [12]. This was related to the target pharmacodynamic index element information available through the levocetirizine pharmacodynamic model setting approach in this study being limited to changes in the sizes of the wheal and flare. Since levocetirizine is not directly involved in the flare and wheal size control response, but is instead an indirect response caused by the antihistamine effect, it was logical to explain it as an indirect response model rather than a direct response model [13]. Similarly, in previous reports [13,15,16], the pharmacodynamics of antihistamines were explained using an indirect response model, which was consistent with the results of this study.

Appendix A show the mathematical information and configuration parameter values of the levocetirizine pharmacodynamic model established in this study, respectively. Regarding the explanation of the wheal and flare size change pattern, the pharmacodynamic model structure for each wheal and flare element was set to be the same, and the model parameter values were optimized through fitting the reported observations. Figure 3 shows the graphical results of fitting the indirect response input inhibition model of the levocetirizine pharmacodynamic data. The reported observations [12] all overlapped within the prediction ranges of the levocetirizine pharmacodynamic model established in this study. In other words, it was confirmed that the established levocetirizine pharmacodynamic model appropriately explains all observations of changes in the wheal and flare sizes [12] according to levocetirizine exposure within the 95% prediction interval. This implied that the pharmacodynamic model established in this study could be used to predict the pharmacodynamic patterns of levocetirizine at a reasonable level according to levocetirizine plasma exposure.

### 3.4. Exploring Gender Difference in Levocetirizine Pharmacometrics

Model simulations were performed through numerical changes and a reflection of the selected effective covariates in the final levocetirizine population pharmacokinetics–pharmacodynamics co-linked model. Valid covariates were gender, body weight, BSA, and albumin levels explored in the levocetirizine population pharmacokinetic modeling process. Gender was treated as a categorical type, and body weight, BSA, and albumin were treated as continuous types. When applying continuous covariates, the median, minimum, maximum, and mean values in the male and female groups, respectively, were reflected. Figure 4 shows the results of the pharmacokinetic and pharmacodynamic simulations after oral administration of the levocetirizine 5 mg tablet, which reflect gender and the body weight, BSA, and albumin levels in that gender. 

According to the simulation of the established levocetirizine population pharmacokinetic-pharmacodynamic model (which reflected all quantitative relationships of the effective covariates explored in the modeling process), the C_max_ and maximum effect were higher, and the elimination from the plasma and recovery to the baseline effect were faster in women than in men. This may have been related to the concentration profile patterns of levocetirizine in plasma for each gender according to levocetirizine exposure. However, in the statistical comparison of the model predictions in the pharmacokinetic and pharmacodynamic aspects, no significance between genders was confirmed (*p* > 0.05), suggesting that the influence of gender on the pharmacokinetic and pharmacodynamic diversity of levocetirizine within the population would not be very high. In the graph (Figure 4), the baseline value refers to the immediate skin reactions that occurred due to exogenous histamine exposure before levocetirizine administration. As the value approaches 0 from the baseline value, the antihistamine effect increases due to levocetirizine exposure. Figure 5 shows the pharmacokinetic and pharmacodynamic simulation results after multiple exposures to the levocetirizine 5 mg tablet, reflecting gender and body weight, BSA, and albumin levels for that gender. 

The same patterns as a single exposure were confirmed in the steady-state outcome predictions for multiple oral exposures to levocetirizine (assuming the general clinical use of once-daily exposure). In other words, the C_max_ and maximum effect were higher in women than in men, and elimination from the plasma and recovery to the baseline effect were faster during the same time period. There was a significant difference between genders (*p* < 0.05) in the flare size changes in a steady state, with the average degree of flare size inhibition (related to antihistamine response) being significantly lower in females than in males. This implied that upon long-term continuous exposure to levocetirizine, significant differences in the antihistamine effect between genders may occur depending on the dosage regimen. In other words, when the single-exposure dose of levocetirizine is increased or the clinical dosage is changed to shorten the administration interval, significant differences may occur in the pharmacokinetic and pharmacodynamic aspects of levocetirizine between genders. 

Table 1 shows a quantitative comparison of plasma exposure and drug efficacy according to gender after single and multiple oral administration of the levocetirizine 5 mg tablet. According to the quantitative comparison of the model simulations reflecting gender and mean body weight, BSA, and albumin levels, there was no large difference in the total plasma exposure (AUC) and the extent of the effect (area under the effect curve, [AUEC]) between genders. Even in the mean values, the absolute differences between genders were within 10 ng/mL (for plasma concentrations) or % (for change in wheal and flare sizes), and the differences were not large. However, fluctuations were greater in women than in men in both pharmacokinetics and pharmacodynamics. This was associated with a higher C_max_ and shorter T_1/2_ and MRT in females than in males, as seen in the pharmacokinetic results (Figure 1 and Appendix A depict the pharmacodynamic simulation results according to changes in the exposure interval of the levocetirizine 5 mg tablet, showing a comparison between genders in the wheal and flare size changes, respectively. Based on the model simulation results, according to the change in the administration interval of the levocetirizine 5 mg tablets, the antihistamine effects (related to changes in the wheal and flare sizes) in both males and females tended to return to the original baseline value as the administration interval increased, and the fluctuations tended to increase. That is, it was predicted that upon multiple exposures to the levocetirizine 5 mg tablets with an administration interval of 12 h, the antihistamine effects would continue to appear at a level close to 60–100% without complete recovery to baseline. On the other hand, upon multiple exposures to the levocetirizine 5 mg tablets with an administration interval of 72 h, the antihistamine effects were completely restored to baseline just before the next administration, and fluctuations were predicted to be significant, up to approximately 65–100%. Like the model prediction results for multiple exposures at 24 h dosing intervals (Figure 5), it was confirmed that there were no large differences between genders in both pharmacokinetic and pharmacodynamic aspects of the levocetirizine 5 mg tablet multiple exposure model simulation results at 12, 48, and 72 h dosing intervals. As a result, it was implied that in the clinical application of levocetirizine 5 mg tablets, considering gender would not be important if there were no special disease factors (such as renal dysfunction) in the target patient group. No significant differences between genders were identified at the 12 and 72 h dosing intervals with respect to the change in flare size (*p* < 0.05), which were associated with high levels of sustained suppression and complete recovery to baseline, respectively. In this study, which demonstrated model simulations, the administration intervals of 24, 48, and 72 h for the levocetirizine 5 mg tablets were established based on the recommended dosage according to the renal function levels in existing clinical information.

## 4. Discussion

The results of this study suggest that the influence of gender would not be very significant when applying the general dosage regimen in terms of the pharmacokinetics and pharmacodynamics of levocetirizine. Despite clear differences in the pharmacokinetics between genders, the degree of change in pharmacokinetics and pharmacodynamics due to gender was relatively small. Even in the model simulation results following multiple exposures to levocetirizine (with various dosing interval changes), no large differences were identified in the mean pharmacokinetic and pharmacodynamic profiles between genders. Therefore, it was suggested that a consideration of gender would not be critical in relation to inter-individual pharmacokinetic and pharmacodynamic diversity in the clinical application of levocetirizine at a general dosage (5 mg/day). To date, there have been no reports of side effects that differ depending on gender from general dosage exposure to levocetirizine. However, as confirmed in this study, the differences in pharmacokinetics and pharmacodynamics between genders at the general dosage of levocetirizine are not likely to be very large. Therefore, it was expected that the general dosage of levocetirizine would be unlikely to cause dramatic differences in efficacy or side effects between genders. In other words, although the degree of difference in sensitivity of levocetirizine response between genders needs to be confirmed in the future, it was judged that the possibility of differences between genders regarding the efficacy or side effects (at the general dosage) of levocetirizine at the current research stage is unlikely to be an important factor to consider in clinical practice. Nevertheless, this study had great significance in that it was able to quantitatively explore the pharmacokinetic and pharmacodynamic characteristics of levocetirizine and their effects across genders, which had not been clearly identified previously. In addition, it is important to accelerate precision medicine and present information related to drug efficacy and safety by discovering new covariates that can effectively explain the diversity of levocetirizine pharmacometrics within the population [14,21]. In particular, the fact that the gender factor was an effective covariate in the distribution of levocetirizine to peripheral tissues was a new and progressive discovery. As a result, it was confirmed that when exposed to the same dose of levocetirizine tablets, the distribution of levocetirizine in peripheral tissues was higher and the degree of antihistamine response tended to be higher in males than in females. These were interesting results that greatly complemented the gaps in existing knowledge about the relevance of gender factors in levocetirizine clinical trials.

Levocetirizine is generally recognized as a drug with a relatively high safety margin, with fewer side effects compared to first-generation antihistamines [4,8]. However, not using caution with the dosage regimen based on the empirical safety of levocetirizine is unscientific and would greatly deviate from the purpose of the clinical application of medicines based on accurate pharmacometric information pursued by precision medicine. From a scientific standpoint, knowing accurate pharmacometric information and applying medicines in clinical practice are very important, and exploring pharmacometric differences between genders, which was the focus of this study, will be a particularly important factor to consider in clinical practice. This is because gender factors inevitably occur in clinical practice and are easily accessible pieces of information, which is important in the application of pharmacotherapy. The confirmation of pharmacometric differences between genders can be used as very important efficacy and safety information in the future development process of improved or new drugs (expanded based on the existing core structure) and/or formulations. In addition, it has become necessary to identify differences in pharmacokinetic characteristics and drug efficacy between genders to clearly indicate the drug’s usage on the label through the approval of drug licensing authorities in the future.

Although levocetirizine is a drug whose dose adjustment is recommended according to renal function [22], typical renal function-related indicators were not explored as effective covariates in this study. This is because the population applied in the modeling was a group of healthy adults with normal renal function, with creatinine clearance (CrCL) and estimated glomerular filtration rate (eGFR) values of 80–190 mL/min and 70–150 mL/min/1.73 m^2^, respectively. According to the recommended dose adjustment information for levocetirizine related to renal function [23,24], if the CrCL is 50–80 mL/min, it is recommended to take the levocetirizine tablet (5 mg) once a day; if the CrCL is 30–50 mL/min, it is recommended to take the levocetirizine tablet (5 mg) once every 2 days; and if the CrCL is 10–30 mL/min, it is recommended to take the levocetirizine tablet (5 mg) once every 3 days. If the eGFR level is 60 mL/min, it is recommended to take the levocetirizine tablet (5 mg) once a day; if the eGFR level is 30–60 mL/min, it is recommended to take the levocetirizine tablet (5 mg) once every 2 days; and if the eGFR level is 15–30 mL/min, it is recommended to take the levocetirizine tablet (5 mg) once every 3 days.

In this study, a calculation of various biochemical parameter values and pharmacokinetic correlation analysis were performed that focused on other indicators of renal-related functions [25] in relation to the interpretation of levocetirizine pharmacokinetic diversity, but the reflection of the covariates of renal function factors was limited. This implied that levocetirizine administration could be applied without considering renal function in a group of patients without significant renal function problems. Although the renal function indices were within normal values, they did not have a significant effect on levocetirizine’s V/F and CL/F, even within relatively wide CrCL and eGFR ranges. In the future, the impact of renal function disease factors on levocetirizine pharmacokinetics will be quantifiable through conducting levocetirizine pharmacokinetic studies that include groups with significantly reduced renal function (such as cases where the CrCL and eGFR are less than 50 and 60 mL/min, respectively).

According to past reports [17,26], levocetirizine has a low metabolism in the body, with more than 85% of the oral dose being excreted unchanged in the urine and feces. Even in the drug safety information for levocetirizine [22], separate dosage adjustments according to liver function are not confirmed. This was consistent with the results in this modeling study in which liver function indicators [27] such as alkaline phosphatase (ALP), aspartate transaminase (AST), alanine transaminase (ALT), and gamma-glutamyl transferase (GGT) were not ultimately explored as valid covariates. Therefore, it is suggested that liver function may not be a major factor in the clinical application of levocetirizine, especially in healthy adult populations. There are no confirmed reports of gender differences in the expression of transporters and metabolic enzymes related to the pharmacokinetics of levocetirizine. However, the differences in the pharmacokinetic analysis results of levocetirizine between genders confirmed in this study were not large enough to require caution in clinical practice. This suggested that potential differences in the expression of levocetirizine transporters and metabolic enzymes would not be a major consideration in interpreting pharmacokinetic variation between genders.

The pharmacodynamics of levocetirizine were explained by the indirect response with the input inhibition model, considering the physiological mechanism of histamine action antagonism by the drug [2,28]. The pharmacokinetic–pharmacodynamic correlation in which levocetirizine selectively inhibits histamine receptors [28], resulting in an antihistamine response (such as wheal and/or flare size changes), was logical enough to be explained by an indirect response model. Considerations of the covariates in constructing the levocetirizine pharmacodynamic model were limited, however. This is related to the obvious limitations in obtaining candidate covariate information and setting targets for the IIV interpretation of the levocetirizine drug response within the current data accessibility range. As antihistamine effects of levocetirizine, changes in wheal and flare sizes on the skin may differ between genders. According to past reports [29,30], there are differences between genders in the thickness of skin tissues and structures, which is suspected to be mainly caused by differences in sex hormones. Although it is reported that there is no significant difference between genders in the density and distribution of mast cells in the skin [30], considering reports of sex hormones that can modulate the thickness of the epidermis and dermis as well as the immune system functions in the body [29], the possibility that gender differences may occur in the pharmacodynamic parameters related to the skin reaction of levocetirizine cannot be completely ruled out. Nevertheless, comparisons of levocetirizine pharmacodynamics between genders in this study could only be performed by predicting results based on gender differences in pharmacokinetics. In the future, it will be necessary to conduct exploratory studies on the occurrence and sensitivity of the levocetirizine response variation between individuals and the effective factors (including gender) associated with it. In addition, this study has limitations in that a pharmacodynamic model expansion was conducted based on the pharmacokinetic and pharmacodynamic results according to levocetirizine exposure in a population aged 6 to 11 years. Therefore, it will be necessary to conduct prospective clinical studies on the pharmacodynamics of levocetirizine in adults.

## 5. Conclusions

In this study, covariate correlations were established for BSA, albumin, and gender in the IIV interpretation of the levocetirizine V_p_/F, and BSA and body weight were explored in relation to the CL_p_/F and K_a5_, respectively. In an effort to investigate the impact of gender-based differences in levocetirizine pharmacokinetics on drug efficacy, we sought to extend the co-linkage of the population pharmacokinetic model to incorporate a pharmacodynamic model. In the comparison between genders, the levocetirizine C_max_ and level of the maximum drug effect were higher in women than in men, and its elimination from the plasma and recovery to baseline were faster during the same time exposure period. However, despite the differences in the levocetirizine pharmacokinetics between genders, the magnitude was not large and not very significant in terms of efficacy. Therefore, it was implied that a gender consideration would not be a critical point when administering 5 mg levocetirizine tablets, which is commonly used in clinical practice, to general patient groups (excluding patients with severe renal function impairment) based on the pharmacokinetic and pharmacodynamic aspects. This study presents a new interpretation of levocetirizine pharmacokinetic variations between genders and the resulting pharmacodynamic changes through a quantitative pharmacometrics approach. In addition, a very useful perspective (particularly related to gender factors) that is further expanded is being proposed in the scientific precision medicine of levocetirizine.

## Figures and Tables

**Figure 1 pharmaceutics-16-00146-f001:**
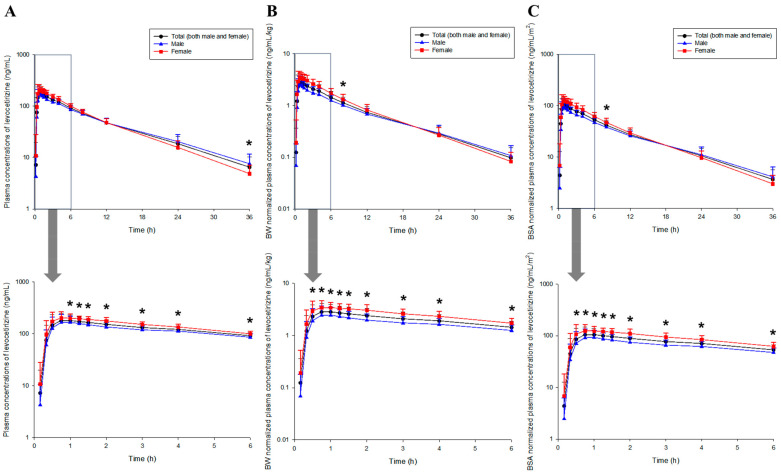
Comparison of pharmacokinetic profiles between genders (24 males and 16 females) following single oral administration of the levocetirizine 5 mg tablet. (**A**–**C**) indicate plots based on raw data, plasma concentration normalized to each individual’s body weight, and plasma concentration normalized to each individual’s body surface area (BSA), respectively. Arrows in the graph indicate zoomed-in concentration profile patterns from 0 to 6 h. In the graph, observations are represented as mean and standard deviation by dots and upper vertical bars, respectively. * *p* < 0.05 compared between males and females.

**Figure 2 pharmaceutics-16-00146-f002:**
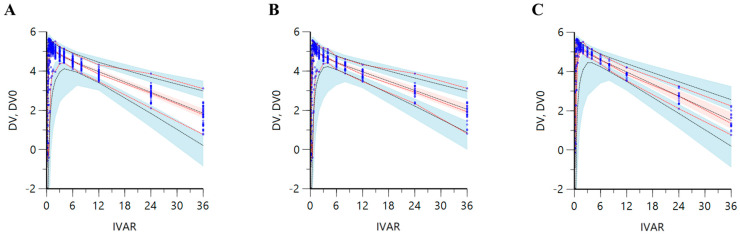
Visual predictive check (VPC) results for the population pharmacokinetic model of observed plasma concentrations (natural log scale, ng/mL) over time following oral administration of levocetirizine 5 mg tablet. The VPC results were displayed for the entire data (**A**) without any stratification, and for each male (**B**) and female (**C**), stratified by gender. Dots represent the observed concentrations. Black dashed lines represent the 97.5th, 50th, and 2.5th percentiles of the predicted concentrations. Red dashed lines represent the 97.5th, 50th, and 2.5th percentiles of observations. Blue shaded regions represent the 95% confidence intervals for the predicted 2.5th and 97.5th percentiles. Red shaded regions represent the 95% confidence intervals for the predicted 50th percentiles. IVAR on the X-axis in the graph is an independent variable, meaning the time after oral administration of levocetirizine, and the Y-axis represents the dependent variable, meaning the observed values (DV) of the levocetirizine concentration in plasma and the predicted values (DV0) from the model.

**Figure 3 pharmaceutics-16-00146-f003:**
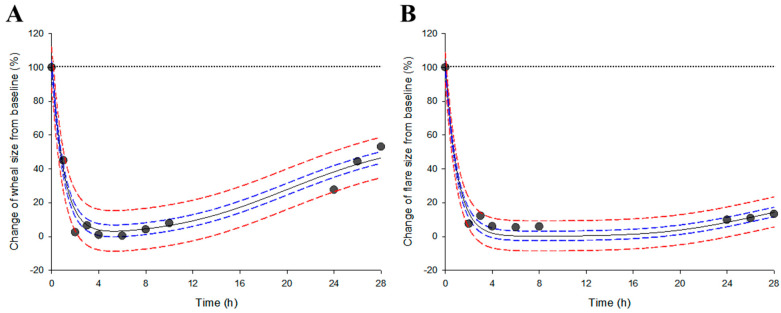
Results of fitted data between the observations and predictions of a pharmacodynamic model established based on the reported pharmacokinetic–pharmacodynamic observations in a population exposed to 5 mg levocetirizine. (**A**,**B**) refer to changes in the wheal and flare sizes as a histamine-induced response and a histamine-suppressed response produced by levocetirizine, respectively. Black solid lines and dots represent mean values based on the model predictions and observed values, respectively. The blue and red dashed lines represent the 95% confidence interval and prediction interval of the mean based on the model predictions, respectively. The black dotted lines in the graph refer to the baseline value of the histamine-induced response.

**Figure 4 pharmaceutics-16-00146-f004:**
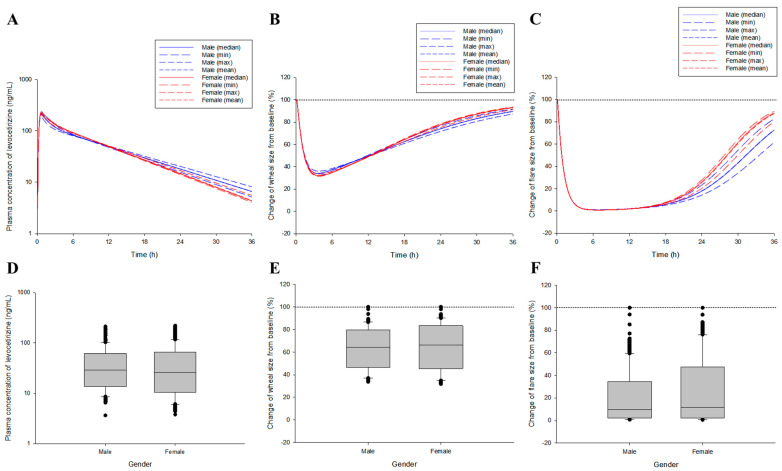
Prediction results of the plasma concentration and drug efficacy (as a change in the wheal and flare sizes) profiles according to single oral exposure to levocetirizine (5 mg) between genders after reflection of covariates in the levocetirizine population pharmacokinetic model. (**A**–**C**) refer to the results estimated by applying the median, minimum (min), maximum (max), and mean biochemical parameter values for each gender to the covariate correlation within the model. (**D**–**F**) refer to a comparison between genders and the results (0 to 36 h) estimated by applying the mean biochemical parameter values in each gender to the covariate correlation within the model. The black dotted lines in the graph (**B**,**C**,**E**,**F**) represent the baseline (as no inhibition) of the histamine (H_1_)-induced response.

**Figure 5 pharmaceutics-16-00146-f005:**
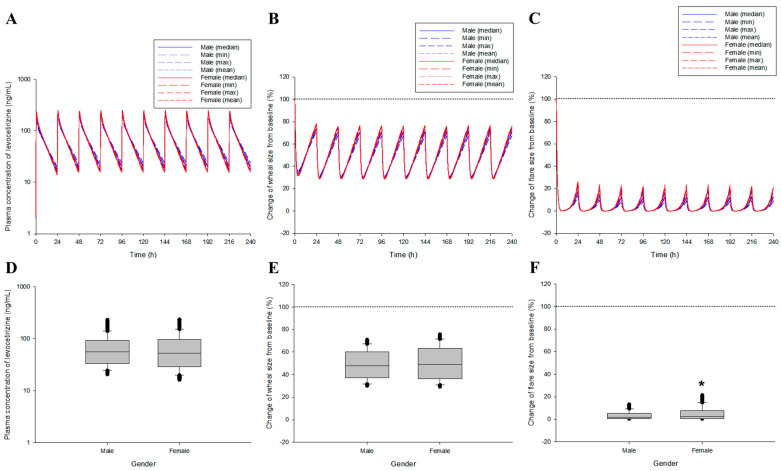
Prediction results of the plasma concentration and drug efficacy (as a change in the wheal and flare sizes) profiles according to multiple oral exposures to levocetirizine (5 mg, 24 h dosing interval) between genders after reflection of the covariates in the levocetirizine population pharmacokinetic model. (**A**–**C**) refer to the results estimated by applying the median, minimum (min), maximum (max), and mean biochemical parameter values for each gender to the covariate correlation within the model. (**D**–**F**) refer to the comparison of the results between genders at steady state (216–240 h) estimated by applying the mean biochemical parameter values in each gender to the covariate correlation within the model. The black dotted lines in the graph (**B**,**C**,**E**,**F**) represent the baseline (as no inhibition) of the histamine (H_1_)-induced response. * *p* < 0.05 compared to the value in males.

**Table 1 pharmaceutics-16-00146-t001:** Prediction results of plasma concentration and drug efficacy in the average group between genders according to single and multiple (24 h dosing interval) oral exposures to levocetirizine (5 mg).

Parameter	Plasma Concentrations (ng/mL)		Change in Wheal Size (%)		Change in Flare Size (%)	
	Single Dose					
	Male	Female	Male	Female	Male	Female
Maximum value (0–36 h)	212.00	220.52	89.45 ^b^	92.77 ^b^	72.56 ^b^	87.14 ^b^
Minimum value (0–36 h)	6.53 ^a^	4.39 ^a^	33.81	31.87	0.77	0.65
Fluctuation ^d^	32.46	50.27	2.65	2.91	93.81	134.14
Mean value (0–36 h)	45.15	46.09	63.02	64.30	20.31	25.93
AUC_all_ (h·ng/mL)	1624.90	1659.09	NA	NA	NA	NA
AUEC (h·%)	NA	NA	1330.75	1284.66	2867.43	2665.73
	**Multiple exposure**					
Maximum value ^c^	231.37	235.37	70.81 ^b^	75.54 ^b^	13.11 ^b^	21.10 ^b^
Minimum value ^c^	20.69 ^a^	16.19 ^a^	30.17	29.39	0.44	0.43
Fluctuation ^d^	11.18	14.53	2.35	2.57	29.92	49.24
Mean value ^c^	70.73	70.71	48.79	50.22	3.49	5.05
AUC_all_ (h·ng/mL) ^c^	1702.49	1702.49	NA	NA	NA	NA
AUEC (h·%) ^c^	NA	NA	2431.14	2397.15	3517.14	3480.30

NA: not applicable. ^a^ The lowest value was defined as the time (T_max_) after reaching the maximum plasma concentration (C_max_). ^b^ The highest value was defined as the time after reaching maximum efficacy (E_max_). ^c^ The values at 216–240 h were taken as the steady state following multiple exposures. ^d^ This was calculated as the ratio between the maximum and minimum values. The data in this table are a result of applying the mean biochemical parameter values in each gender to the covariate correlation within the model. AUEC refers to the area under the curve based on the baseline in the drug efficacy–time curve. The change values in the sizes of the wheal and flare were presented based on 100 as the baseline (immediate skin reaction caused by exogenous histamine exposure before levocetirizine administration), and the closer the values were to 100 or 0, the lower and higher the antihistamine effect according to exposure to levocetirizine, respectively.

## Data Availability

All data and related materials are accessible in this manuscript and the Appendix A.

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
