# Peer review of "Is Gender an Important Factor in the Precision Medicine Approach to Levocetirizine?"

_pharmaceutics, 2024, doi:10.3390/pharmaceutics16010146_

Round 1

Reviewer 1 Report

Comments and Suggestions for Authors

The research study focuses on gender differences in levocetirizine pharmacokinetics and to quantitatively predict and compare the effects of the pharmacokinetic differences between genders on pharmacodynamics through population pharmacokinetic-pharmacodynamic modeling investigated in manuscript "Is gender an important factor in the precision medicine approach to levocetirizine?". The word is performed and presented with a systematic way. Few comments are as follows:

1. Is there any reported side effect in females when given with normal male dose? 

2. Is previous such reported data there of varion of PK PD for this drug based on gender? 

3. Was The study performed in female subjects while excluding any particular criteria (like menstruation duration etc)? Similarly for male. 

4. Is the study performed as per ethical guidelines? 

Author Response

We greatly appreciate Reviewer 1's interest in our study and manuscript.

Please check the attached “Point-by-point response” file for accurate and detailed comments and related answers.

We have attached a "Point-by-point response" file written in Word file.

Reviewer 2 Report

Comments and Suggestions for Authors

Jeong et al. reported the impact of sex-dependent differences on the PK of levocetirizine. The author mentioned that the findings from the study will help to narrow the knowledge gap related to scientific precision medicine by presenting new previously unconfirmed 31 information on gender differences in levocetirizine pharmacometrics. Please, see below my comments on this manuscript.

1.       It is very unclear in the introduction why levocetirizine was used to establish a gender-specific PK/PD relationship. Was there more evidence of sex differences in the PK/PD of levocetirizine?

2.       I also didn’t see enough information on the disposition of levocetirizine in the manuscript to understand if there is any impact of sex differences in the expression of drug-metabolizing enzymes and transporters on the PK of levocetirizine.

3.       Also, the author extrapolated the PD model of ages 6 to 11 years individuals in the adult population considering that there are age-dependent differences in the population. It is good to cite if any references are reported on the same.

4.       Please, proofread the manuscript again before submission.

5.       Please, make the referencing style consistent. 

Author Response

We greatly appreciate Reviewer 2's interest in our study and manuscript.

Please check the attached “Point-by-point response” file for accurate and detailed comments and related answers.

We have attached a "Point-by-point response" file written in Word file.

Reviewer 3 Report

Comments and Suggestions for Authors

This paper examines the pharmacokinetic and pharmacodynamic variability by gender in establishing the clinical use of levocetirizine. Information on interindividual variability in levocetirizine pharmacokinetics is still lacking. In particular, the extent of pharmacokinetic and pharmacodynamic variability by gender has not been clarified. The purpose of this study was to explore gender differences in the pharmacokinetics of levocetirizine. Population pharmacokinetic-pharmacodynamic modeling will be used to quantitatively predict and compare the impact of sex differences in pharmacokinetics on pharmacodynamics. This study presents new, previously unidentified information on sex differences in levocetirizine pharmacokinetics. This will be useful in closing the knowledge gap associated with scientific precision medicine.

In this manuscript, data analyses and analyses are generally done well. The elucidated results and conclusions sound reliable. Discussions are well described. Therefore, I may suggest publication of this work in Pharmaceutics from scientific viewpoints. However, construction of this manuscript is not well prepared for non-specialist readers. It may not be well understandable at instant reading. In order to fix this point, addition of (i) the initial figure to explain research outline and (ii) the final figure to explain conclusion statements are recommended.

Author Response

We greatly appreciate Reviewer 3's interest in our study and manuscript.

Please check the attached “Point-by-point response” file for accurate and detailed comments and related answers.

We have attached a "Point-by-point response" file written in Word file.
